# Ungulates’ Behavioral Responses to Humans as an Apex Predator in a Hunting-Prohibited Area of China

**DOI:** 10.3390/ani13050845

**Published:** 2023-02-25

**Authors:** Mingzhang Liu, William J. McShea, Yidan Wang, Fan Xia, Xiaoli Shen, Sheng Li

**Affiliations:** 1School of Life Sciences, Institute of Ecology, Peking University, Beijing 100871, China; 2Conservation Ecology Center, Smithsonian Conservation Biology Institute, Front Royal, VA 22630, USA; 3State Key Laboratory of Vegetation and Environmental Change, Institute of Botany, Chinese Academy of Sciences, Beijing 100093, China

**Keywords:** human disturbance, fear ecology, acoustic cues, Automated Behavioral Response (ABR) system, playback experiment

## Abstract

**Simple Summary:**

Large mammals’ behavioral responses to humans as predators may be impacted by hunting intensity. Using a playback experiment, we found that two wild ungulates exhibited reactive (flee) rather than proactive responses (decrease habitat use) to human vocalizations at a hunting-prohibited site in North China. The wild ungulates had equal or even higher flight probabilities upon hearing vocalizations of humans than the native extant large carnivore (leopard). We also found habituation-type responses featured as progressively decreased responses to the vocalizations in both ungulates.

**Abstract:**

Large mammals can perceive humans as predators and therefore adjust their behavior to achieve coexistence with humans. However, lack of research at sites with low hunting intensity limits our understanding of how behavioral responses of animals adapt to different predation risks by humans. At Heshun County in North China, where hunting has been banned for over three decades and only low-intensity poaching exists, we exposed two large ungulates (Siberian roe deer *Capreolus pygarus* and wild boar *Sus scrofa*) to the sounds of humans, an extant predator (leopard *Panthera pardus*) and a control (wind), and examined their flight responses and detection probabilities when hearing different type of sounds. Both species showed higher flight probabilities when hearing human vocalization than wind, and wild boar were even more likely to flee upon hearing human vocalization than leopard roar, suggesting the behavioral response to humans can equal or exceed that of large carnivores in these two ungulates even in an area without hunting practices. Recorded sounds had no effect on detection probability of both ungulates. Additionally, with repeated exposure to sounds, regardless of treatment, roe deer were less likely to flee and wild boars were more likely to be detected, indicating a habituation-type response to sound stimuli. We speculate that the immediate flight behavior rather than shifts in habitat use of the two species reflect the low hunting/poaching pressure at our study site and suggest further examination of physiological status and demographic dynamics of the study species to understand human influence on their long-term persistence.

## 1. Introduction

Humans can function as an apex predator in ecosystems, killing wildlife at rates that equal or exceed that of non-human predators [1]. Fear of humans is thought to be an important mechanism mediating human effects on wildlife behavior, including enhanced vigilance [2], reduction of animal movements [3] and reduced activities in daylight [4]. These changes in wildlife behavior have the potential of influencing the community structures and ecological functions within ecosystems through a trophic cascade [5,6,7]. Despite a large body of evidence demonstrating humans’ negative effects on wildlife habitat use [8,9,10], diurnal activity [11,12] and foraging intensity [2,5,13], whether these effects result from animals’ fearful response to humans as predators or response to general disturbances (e.g., sudden noise, quick-approaching object) still requires examination [6,14,15].

Playback experiments, which expose animals to acoustic cues, provide a powerful and relatively easily-implemented method to investigate inter-specific interactions by quantifying an animal’s response to acoustic stimuli [16,17]. Playback experiments have been utilized to explore anti-predator responses of prey species, especially ungulate responses to large carnivores [18], and also wildlife responses to humans [19,20]. Recent experiments have incorporated camera traps, which automatically record animal behavior, into playback experiments to overcome the potential bias induced by the presence of the observer in the traditional methods [16,17]. This integrated method has been implemented on several mammal species including *Puma concolor* [6,21], *Meles meles* [22], *Lynx rufus*, *Mephitis mephitis*, and *Didelphis virginiana* [6,23], *Odocoileus virginianus* [24], and *Alces alces* [25]. These studies have documented behavioral responses usually associated with fear of a predator, and substantiate that the presence of humans can mediate wildlife behavioral changes by creating a landscape of fear [26].

The behavioral responses of wildlife to human activities can be highly variable [27], implying that the strength of the response is context-dependent. Animals’ perceived risks from humans may be correlated to their probabilities of being killed by humans, as hunting can intensify their responses to human activities [15]. For example, brown bears (*Ursus arctos*) in Scandinavia reduced diurnal activities only after the start of annual bear hunting season [11], and African elephants (*Loxodonta africana*) exhibit stronger responses to voices of adult men than women and youth [19]. Animals may be able to assess the lethality of human activities and reduce their intensity of response to non-lethal activities. Caribou (*Rangifer tarandus*) in Svalbard reduced flight distances with increased number of approaches by the researcher [28], and flight probability of guanacos (*Lama guanicoe*) near the roads decreased progressively once poaching ceased in Argentina [29]. Animals’ ability to assess predation risks from humans can help them cope with human encounters, and avoid excessive costs of unnecessary behavioral responses [30]. However, experimental tests of wildlife fear responses to humans have been largely limited to regions where human hunting is common (e.g., Suraci et al. [6], for mesocarnivores in Santa Cruz Mountains under predator control; Crawford et al. [24], for white-tailed deer in southwestern Georgia as game species), and studies in areas with low hunting intensity where wildlife may respond to humans differently are lacking.

To document whether wildlife perceive humans as predators at sites with low hunting intensity, and how they behaviorally respond to human presence, we conducted a playback experiment by exposing two large ungulates (Siberian roe deer *Capreolus pygargus* and wild boar *Sus scrofa*) to sounds of humans, an extant large carnivore (leopard *Panthera pardus*), and wind as a control sound. Hunting has been banned for over three decades since the enactment of the Wildlife Protection Law, but poaching still exists in some parts of China. Our study site in Shanxi Province supports a single native large carnivore, the leopard, with a relatively high density (4.23 individuals/100 km^2^, 95% CI 2.82–5.64; [31]) compared to leopard populations in other parts of China. The landscapes here are also highly modified by human activities and densely occupied by humans, with poaching activity of low intensity. Both roe deer and wild boar are common (occupancy > 0.9) and are prey of leopards [32], showing no spatial avoidance to sites frequented by humans [33]. Based on camera trap data recorded during the playback experiment, we examined the flight behaviors and detection probabilities of the two ungulates in response to continued broadcasting of the three types of sounds. We compared the relative strength of the behavioral responses of the two species to different sound stimuli and their gradual changes as the experiment progressed. We hypothesized that even at a very low poaching intensity and a high leopard density, fear of humans may outweigh fear of leopards in both ungulate species.

## 2. Materials and Methods

### 2.1. Study System

We conducted this study in a temperate forest in Heshun County of Shanxi Province, China. Located in the central Taihang Montains of North China, the study area is surrounded by highly modified landscape (Figure 1a; [34]). This area is characterized by rolling hills with elevation ranging from 1200–1700 m, and is mainly covered by secondary mixed forest consisting of coniferous and broad-leaved deciduous tree species. About 2000 residents from 9 villages live in this area, and engage in farming, cattle raising, and collecting of non-timber forest products (e.g., medical herbs, mushrooms, etc.). Residents from the county town (about 15 km from the study site with a total population about 55,000) also frequently visit this area. The poaching intensity is low, with less than 20 detections of hunters (e.g., people with shotgun or hunting dogs) recorded per year during the camera-trapping survey from 2017–2019 in a more extensive area encompassing our study site (56,000 camera-days from 123 sites in over 1500 km^2^) [35].

### 2.2. Playback Experiment

We used an Automated Behavioral Response (ABR) system [16,17] to measure the behavioral response of ungulates to a suite of sounds present within the landscape (e.g., human, leopard, wind; Figure 1b). The ABR system is composed of one infrared-triggered camera and one waterproof loudspeaker, powered by a 10,000 mAH lithium battery and a solar panel. The loudspeaker is connected to the camera and can be triggered by the heat sensor of the camera. Thereby, warm-blooded animals passing by the ABR trigger the heat sensor of the camera, and the speaker then plays a random 30-s sound clipping that is stored in the ABR while the camera simultaneously records a video of the same length. We extracted 8 clippings for each sound type (e.g., the sound of man reading, leopard roaring, or wind blowing) from videos downloaded from YouTube (www.youtube.com [accessed on 5 December 2021]; searched by “Chinese reading” in Chinese, “leopard roaring” and “wind blowing”, respectively) and edited all for consistency of amplitude and clipped them to 30 s. The sound clips were broadcasted at a consistent mean sound pressure of 70 dB (measured at 1 m from the ABR system using a decibel meter app on a iPhone 12 [Apple Inc., Copertino, CA, USA]), following the settings used in previous studies [6,23,24].

We selected 36 experimental sites in mixed forests and placed one ABR (supplied by Qingdao Yequ Nature Technology Co., Ltd., Qingdao, China) at each site (Figure 1c). Each ABR was attached to a tree at height of 60–80 cm above ground along the trail to maximum the detection of the two study species. Before being set, each ABR was tested to assure that they could be triggered and broadcasted the sound clips at the same amplitude. To eliminate the potential bias induced by variation among sites, we employed a repeated-measure design following Suraci et al. [6]. The 36 sites were evenly split into three groups, each group receiving a different type of treatment (human, leopard or wind) in one experimental period, and switched to the other two treatments for the next two experimental periods. Thus, three types of sounds were broadcasted at all sites by the conclusion of the three experimental periods, while the sounds played at each group of sites differed during the same experimental period. Sites from different treatments were separated by at least 1 km to avoid mutual interference. The average home range of roe deer and wild boar at similar sites were estimated (95% minimum convex polygons) as 1.9 km^2^ and 1.7 km^2^, respectively [36,37]. Thus, site separation of 1 km also reduced the probability of the same individual being subjected to different sound treatments in the same experimental period. Within a treatment group, cameras were separated by at least 250 m. The experiment was conducted from December 2021 to June 2022, with a variable time length among experimental periods (40, 82 and 83 days for period 1, 2 and 3, respectively) due to COVID-19 restrictions.

### 2.3. Statistical Analysis

We used a logistic regression model with mixed effects to examine the effects of different types of recorded sounds on the immediate behavior and habitat use of the study species during the study period. For the immediate behavior, we measured the flight probability of the individual after hearing the sound. We grouped detections of the same species at the same site within 30 min into one independent detection. For each independent detection, we determined whether the ungulate fled away from the experimental site (denoted as 1 and 0, respectively) in response to the broadcast. A social group was defined as fleeing if any individual demonstrated a flight response.

We measured the habitat use of the study species by dividing the experiment period into sequential one-week periods from each site when the ABR system was operational (a site-week, hereafter) and assessed whether roe deer or wild boar were detected or not in each site-week (denoted as 1 and 0, respectively). We hypothesized that repeated exposure to the recordings would alter a species’ use of the experimental site, leading to changes in their overall detection probability. We used the logistic regression for our analysis as formal occupancy model did not meet convergence due to the high naïve occupancy rate of the study species (1.00 and 0.97 for roe deer and wild boar, respectively). For our dependent variable we used detection probability during each site-week, rather than the count of independent detections, as the study species were rarely detected more than once during a site-week.

For both analyses (i.e., flight probability and detection probability), we incorporated three variables as fixed effects, including type of recorded sounds (Treatment), number of days (for flight response) or weeks (for site-week detection probability) from the beginning of each experimental period (Length), and the ID of the experimental period (Period). We used leopard call rather than wind sound as the reference level for Treatment to compare whether human effects differed from leopard effects. As the univariate additive models (GAMM) indicated no nonlinear correlation between Length and the probability of flight and detection (estimated degree of freedom = 1), we did not incorporate any higher order of terms for Length in the subsequent analysis. For each analysis, we formulated five candidate models, each with a different hypothesis (Table 1), and also included one model incorporating the interaction term between Treatment and Period to examine whether the potential effect of the treatment varied across experimental periods. Following Crawford et al. [24], the experimental site was used as the random effect for each candidate model to account for potential correlations among independent detections or site-week detections at the same site.

We fitted all candidate models for both flight and detection probability in the two study species separately using the package “lme4” [38] in the R environment [39] and ranked them with Akaike information criteria (AICc) [40,41]. Models with △AICc < 2 were considered as top models with similar performance, and we defined the top model with least variables as the best model following the principle of parsimony [40,41]. We validated the predictive capacity of the models using area under the curve (AUC), and examined potential spatial and temporal autocorrelation of the Pearson residuals using the graphical diagnostic for autocorrelation function, following Zuur et al. [42].

## 3. Results

Three ABR units was lost during the experiment (two in period 2, one in period 3), and we also excluded records when the ABR unit malfunctioned during the site-week. In total, we obtained data from 5909 camera-days in which the ABR was operational (36, 34, and 33 operational sites with 1287, 2665 and 1957 camera-days in period 1, 2 and 3, respectively). During the entire study period we obtained 255 and 224 independent detections for roe deer and wild boar, respectively.

For the flight response, roe deer were more likely to flee upon hearing sounds of leopards than wind (β ± se = −1.52± 0.37, *p* < 0.001), and exhibited a similar probability of flight upon hearing sounds of human and leopards (β ± se = 0.72 ± 0.42, *p* = 0.083) (Table 2; Figure 2). The flight probability of roe deer reduced as the experiment progressed (β ± se = −2.21 ± 0.58, *p* < 0.001) (Figure 2). For wild boar, three top models were considered equivalent (△AIC < 2), with the best model incorporating only Treatment (Model 1) indicating that wild boar were more likely to flee upon hearing human sounds than the sounds of a leopard (β ± se = 0.94 ± 0.43, *p* = 0.031), and also were more likely to flee upon hearing sounds of a leopard than the sounds of wind (β ± se = −1.36 ± 0.43, *p* = 0.002) (Figure 2).

For shifts in habitat use during the course of the study, no model was a better predictor of detection of roe deer than the null model. Model 2 had similar performance to the null model (△AIC = 1.87), and indicated that Length had no significant effect (*p* = 0.57; Table 3; Figure 3). For wild boar, the best model (model 2) indicated that their detection probability increased as the experiment progressed (β ± se = 1.00 ± 0.36, *p* = 0.006) (Table 3; Figure 3). Model 3 that incorporated Treatment had similar performance with the best model, but the effect of treatment was not significant (*p* = 0.20 for human, *p* = 0.49 for wind).

The experimental period and its interaction term with treatment had no significant effect on either the flight or detection probability of roe deer and wild boar (Table 2 and Table 3), indicating that the results were stable among the different periods. AUCs indicated adequate predictive capacity of the best logistic regression models (ranging from 0.71 to 0.84), and we found no significant pattern in the auto-correlation functions for the residuals (Appendix A), indicating no evidence for spatial or temporal autocorrelation in the best models.

## 4. Discussion

Despite numerous studies focusing on human effects on ungulate spatio-temporal behaviors, as well as vigilance and foraging intensity [2,5,8,13,43], experimental evidence for their fear and behavioral responses to humans as predators is rare (but see Crawford et al. [24], Bhardwaj et al. [25] and Widén et al. [44]). Our results showed that both ungulates had higher flight probabilities upon hearing human sounds than wind, suggesting that the two species exhibit anti-predator responses and actively avoid encounters with humans in our study area. These results complement the hypothesis that presence of humans can create a landscape of fear in some components of the mammal community, manifested in widespread behavioral changes [6,45]. We also found that roe deer and wild boar had similar or higher flight probabilities upon hearing human than leopard sounds at our study site. This result indicates that humans can trigger equal or even stronger behavioral responses in the two ungulates than large carnivores even with low levels of hunting, similar to the ability of large carnivores to induce significant anti-predator behaviors at low population densities [46,47].

Our study also showed that flight response to human cues was a more sensitive response than spatial avoidance of humans as we found no variation in site use between treatments for the two ungulates, which is also consistent with our previous findings that roe deer and wild boar exhibit no spatial avoidance to sites frequented by humans but avoid direct encounters with humans in our study area [33]. Anti-predator behaviors that animals exhibit are results of tradeoffs between energy expenditure, resource acquisition and predation risks [48]. Shifts in habitat use come at the cost of giving up potential resources from suitable habitats (e.g., food), so they are weighed against the benefits of giving up human-occupied sites to reduce human-caused mortalities [49,50]. Previous studies have shown that in areas where humans are the leading predator, such as Scandinavia and Southwest Georgia, U.S.A., ungulates significantly reduce the use of sites with human sound broadcasting [24,25]. At our study site where hunting intensity was low, flight response seems to be a more cost-efficient behavior in response to human cues, allowing the ungulates to use human-frequented sites while reducing their probability of being hunted. Combined with findings from previous experiments in areas with hunting practices, our results demonstrate the ability of ungulate species to assess the predation risks by humans and respond accordingly, which may be a pivotal trait of species to sustain themselves in human-dominated landscapes.

Roe deer and wild boar also exhibited increased flight probability when hearing leopard sounds as compared to hearing wind sounds. Moreover, wild boar were more likely to flee when they heard human sounds than when they heard leopard sounds, while roe deer exhibited similar flight probabilities when they heard both sounds. Since there was no evidence of different hunting pressure by poachers on these two species in our study area, we speculate that this difference was due to wild boar experiencing higher hunting pressure from humans than leopards, as wild boars are not a preferred prey of leopards compared to roe deer [51]. Additionally, although the amplitude of the sound clips were controlled in the experiment, the perception of the sounds could still differ between roe deer and wild boar because of their different frequency range of hearing, which may also lead to bias in the comparison between behavioral responses of roe deer and wild boar. Neither of the two ungulates avoided sites with leopard sounds. This is contrast to our previous studies based on camera-trap data showing that roe deer avoid sites frequented by leopards [33], possibly due to the lack of synergy between sounds and other cues (e.g., olfactory and visual) in this experiment as ungulates rely on multiple senses to detect predators [52]. The potential synergy and relative importance of different types of cues (audio, olfactory and visual) in triggering prey’s behavioral responses to predators is still unclear, and further study examining ungulates’ responses to different combinations of cues would help to fill this gap. Both ungulates fled when hearing wind sounds, although the flight probability was lower than when hearing human and leopard sounds. Similar results (i.e., behavioral responses to sounds as control) have been widely found in previous playback experiments [24,25], which may be due to the suddenness of the sound appearance, though it does not convey any risk-related information. Similarly to a previous experiment [25], our study also showed reduced responses with repeated exposure to the playback. These habituation-type responses in the two ungulates may be due to increased endurance to the sounds or the expulsion of low-tolerant individuals [53]. The use of predator cues has been used as a proxy for predator presence to examine cascading effects mediated by anti-predator behaviors [54] and to reduce human-wildlife conflict [44]. Our results emphasized the necessity of accounting for habituation in long-term experiments aimed at quantifying demographic changes at population level or wildlife management through the use of predator cues.

## 5. Conclusions

As human disturbances within ecosystems increase [34,55], understanding the underlying mechanisms that mediate wildlife behavioral responses is pivotal to achieving human-wildlife co-existence. Our results suggest that human acoustic cues triggered flight responses, but did not change spatial use in our two study ungulate species that shared habitats with humans, which is different from previous findings in areas where the human is a major predator. Our previous study has also shown that roe deer and wild boar exhibit spatial-temporal segregation to avoid direct encounters with humans [33]. These changes in behavior could be a reflection of the low-intensity hunting activities in our study area; whether they affect the fitness of either animal is unclear. Direct comparison of how wildlife perceive risks from humans in areas with and without hunting activities could help to understand the potential mechanisms that mediate the human effect on wildlife behaviors. We suggest further research that integrates an examination of behavioral responses, physiological status, and demographic dynamics of wildlife persisting in shared habitats with humans to understand human impacts on their future sustainability.

## Figures and Tables

**Figure 1 animals-13-00845-f001:**
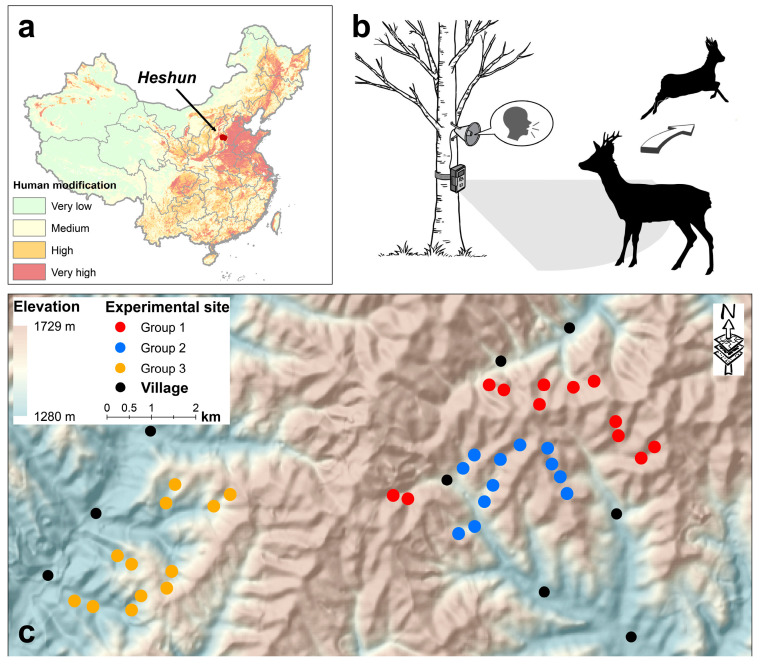
Location of the study area (**a**), experimental procedure (**b**) and the 36 experimental sites of three groups within the study area (**c**). Human modification index in Figure 1 (**a**) (obtained from Kennedy et al. [34]) ranges from 0 to 1 (very low: 0–0.1; moderate: 0.1–0.4; high: 0.4–0.7; very high: 0.7–1.0).

**Figure 2 animals-13-00845-f002:**
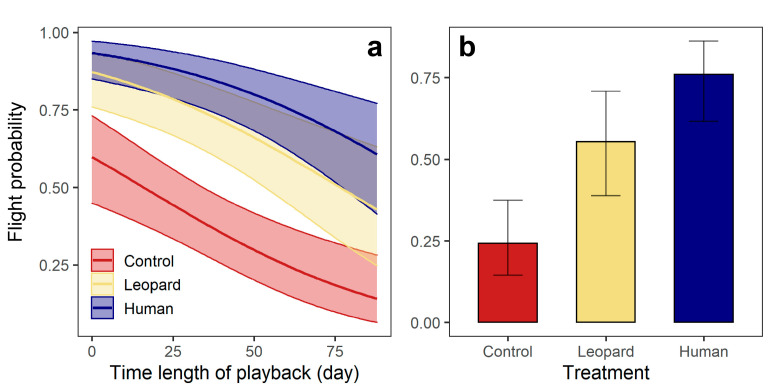
(**a**) Roe deer: flight probability across treatments and correlations between flight probability and time length of experiment; (**b**) wild boar: flight probability across treatment. Shaded areas and vertical bars indicate the 95% confidence intervals of the predicted values.

**Figure 3 animals-13-00845-f003:**
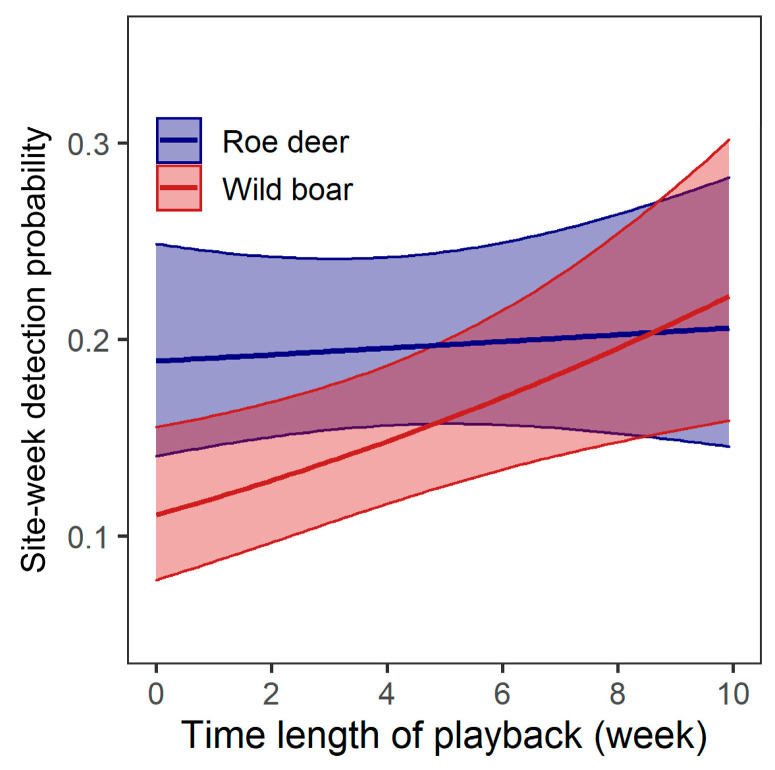
Correlations between occurrence probability of roe deer and wild boar and time length of experiment. Predicted values were obtained by model 2 for roe deer which had comparable performance to the null model (the best model), but indicating a non-significant effect of Length on detection probability. Shaded areas indicate the 95% confidence intervals for the predicted values.

**Table 1 animals-13-00845-t001:** Formula and corresponding hypothesis for the candidate model.

ID	Formula	Hypothesis
1	~Treatment	Ungulates exhibit different behavior in response to sounds of human, leopard and wind.
2	~Length	Ungulates exhibit progressive changes in their intensity of response to the sounds with increased exposure.
3	~Treatment + Length	1 + 2; The intensity of the response does not vary among type of sounds with increased exposure.
4	~Treatment × Length	1 + 2; The intensity of response varies among type of sounds with increased exposure.
5	~Treatment × Period	1; The intensity of response varies among experimental periods.
6	~1	Null model

**Table 2 animals-13-00845-t002:** Rank of logistic regression models for flight probability of roe deer and wild boar based on AICc.

ID	Formula	df	logLik	AICc	Delta	Weight
Roe deer						
3	Treatment + Length	5	−141.51	293.25	0.00	0.80
4	Treatment × Length	7	−140.81	296.07	2.81	0.20
1	Treatment	4	−149.69	307.54	14.28	0.00
5	Treatment × Period	10	−146.53	313.96	20.70	0.00
2	Length	3	−162.40	330.89	37.63	0.00
6	Null	2	−168.05	340.14	46.88	0.00
Wild boar						
1	Treatment	4	−128.90	265.99	0.00	0.46
4	Treatment × Length	7	−126.30	267.11	1.13	0.26
3	Treatment + Length	5	−128.74	267.76	1.77	0.19
5	Treatment × Period	10	−124.16	269.35	3.36	0.09
6	Null	2	−144.16	292.37	26.39	0.00
2	Length	3	−144.16	294.42	28.43	0.00

**Table 3 animals-13-00845-t003:** Rank of logistic regression models for detection probability of roe deer and wild boar based on AICc. The best model is shown in bold.

		df	logLik	AICc	Delta	Weight
Roe deer						
6	Null	2	−412.02	828.06	0.00	0.53
2	Length	3	−411.95	829.93	1.87	0.21
1	Treatment	4	−411.24	830.52	2.47	0.15
3	Treatment + Length	5	−411.13	832.34	4.28	0.06
5	Treatment × Period	10	−406.59	833.46	5.41	0.04
4	Treatment × Length	7	−410.58	835.30	7.24	0.01
Wild boar						
2	Length	3	−354.57	715.18	0.00	0.59
3	Treatment + Length	5	−353.34	716.76	1.58	0.27
4	Treatment × Length	7	−352.43	718.99	3.82	0.09
6	Null	2	−358.36	720.73	5.55	0.04
1	Treatment	4	−357.25	722.55	7.37	0.01
5	Treatment × Period	10	−353.07	726.42	11.24	0.00

## Data Availability

The dataset used for generating this analysis is openly available in ResearchGate (DOI: 10.13140/RG.2.2.13631.38560; available at: https://www.researchgate.net/publication/367253679_Playback_Experiment_Behavior_Records; upload on 21 January 2023). For details about the dataset, please contact M.L. at liumingzhang@pku.edu.cn.

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
