# Peer review of "Ungulates’ Behavioral Responses to Humans as an Apex Predator in a Hunting-Prohibited Area of China"

_animals, 2023, doi:10.3390/ani13050845_

Round 1

Reviewer 1 Report

Ungulates’ Behavioral Responses to Humans as an Apex 2
Predator in Hunting-Prohibited Area of China

Review 30/01/2023

Line 74 ‘are ceased’

Line 123: was male or female reading chosen? And do you think that choice could have been influential?

This study was well designed and considered. The write up of these results was well contextualized in the research. I have no further comments or concerns about this study, and appreciated the chance to review.

Cheers

Author Response

Review 30/01/2023

Thank you very much for your valuable comments on our manuscript! Please find the replies to the comments below.

Line 74 ‘are ceased’

>>> I have checked the original draft, and in Line 74, it is indeed “are ceased”.

Line 123: was male or female reading chosen? And do you think that choice could have been influential?

>>> Thanks for pointing this out! We used clips of reading sounds of male in the experiment, and I have made a revision in Line 123 to specify this. According to a pilot survey, most of the local residents detected by the camera trap are male. Thereby, it is possible that the male and female voice of humans have different impact on the animals’ behavioral responses. However, as our study focus on the ungulates’ response to humans as a potential predator, rather than the ungulates’ ability to discriminate the threats from men and women, we decided to use only the sounds of man.

This study was well designed and considered. The write up of these results was well contextualized in the research. I have no further comments or concerns about this study, and appreciated the chance to review.

Cheers

Reviewer 2 Report

The study answers the question of how they respond two large ungulates (Siberian roe deer Capreolus  pygarus and wild boar Sus scrofa) to the sounds of humans, an extant predator (leopard Panthera pardus) and a control (wind), and examined their flight responses and detection probabilities when hearing different type of sounds. I find the subject original and appropriate for Animals magazine. It fills a gap in the field of wildlife response
for different sounds. There is a lack of research on how animals' behavioral responses adapt to different threats from predators and humans. Authors should write whether they had permission from the relevant authorities to disturb protected species (the sound clips were broadcasted at a consistent mean sound pres sure of 70 dB). The conclusions are consistent with the evidence presented and they answer the main question asked.

Author Response

The study answers the question of how they respond two large ungulates (Siberian roe deer Capreolus  pygarus and wild boar Sus scrofa) to the sounds of humans, an extant predator (leopard Panthera pardus) and a control (wind), and examined their flight responses and detection probabilities when hearing different type of sounds. I find the subject original and appropriate for Animals magazine. It fills a gap in the field of wildlife response
for different sounds. There is a lack of research on how animals' behavioral responses adapt to different threats from predators and humans. Authors should write whether they had permission from the relevant authorities to disturb protected species (the sound clips were broadcasted at a consistent mean sound pres sure of 70 dB). The conclusions are consistent with the evidence presented and they answer the main question asked.

>>> Thank you for your valuable comments on our manuscript! For the permission of the experiment, both roe deer and wild boar are not listed as protected species in China, and because the experiments are non-invasive and are produced outside of the protected area where the actual human disturbances is very prevalent, we think that there is no need for a permission from the relevant authorities.

Reviewer 3 Report

Comments

-        60: Alces alces (Italic !)

-        85, 174-175, 281-282:  Isn't there a possibility that the reaction of the two analyzed species is more obvious in the case of the combination of the leopard's hearing and its sense of smell? As a rule, or often, prey animals react when they smell the predator or when they see it. Therefore, I think that it will not be insufficient for them to hear the roar. At the very least, the sound-only reaction might be irrelevant or non-existent in some situations. Could the sound be combined with the smell in the experiment?    

 Suggestions

1.      Perhaps a comparison between the reactions of the analyzed species during the period of hunting and now (after decades of prohibition) would have been desirable. Of course, the experiment cannot take place now, in the studied area, but maybe elsewhere.  

Author Response

Comments

Thank you for your valuable comments on our manuscript! Please find the replies to the comments below.

60: Alces alces (Italic !)

>>> Thanks for pointing this out and this has been revised.

-        85, 174-175, 281-282:  Isn't there a possibility that the reaction of the two analyzed species is more obvious in the case of the combination of the leopard's hearing and its sense of smell? As a rule, or often, prey animals react when they smell the predator or when they see it. Therefore, I think that it will not be insufficient for them to hear the roar. At the very least, the sound-only reaction might be irrelevant or non-existent in some situations. Could the sound be combined with the smell in the experiment?    

>>> We totally agreed that the synergy between different cues from the predators are important for the prey to assess the predation risks. We have added some further discussion in the manuscript to emphasize this (see Line 283 - 286). However, the sound of the predator only were also widely used to understand the predator-prey interactions (Hettena et al., 2014, Prey responses to predator’s sounds: A review and empirical study, Ethology), and is much easier to handle in a control experiment than olfactory cues (e.g., the olfactory cues may decay with time). Thereby, only sounds were used in our current experiment. In the further research in our study system, we would like to combine different types of predator cues to examine their synergy and compare their relative importance in triggering the prey’s behavioral responses.

 Suggestions

  1. Perhaps a comparison between the reactions of the analyzed species during the period of hunting and now (after decades of prohibition) would have been desirable. Of course, the experiment cannot take place now, in the studied area, but maybe elsewhere.

>>> Thanks for this valuable suggestion! We also feel that it is very important to experimentally compare the human fear effect on wildlife in areas with different hunting pressures, and we added some lines to emphasize this in discussion (see Line 307-309), but such experiment can be difficult to produce in China due to the prohibition of wildlife hunting. European or North America countries would be more appropriate for this kind of study.